# SURFBoard: Reproducible Performance Analysis for Distributed Machine Learning Workflows

## Abstract

Large-scale HPC infrastructures are enablers for scientific research in many domains. The recent advances in machine learning (ML) have led to an ever increasing demand for computation power, as well as the design of complex operational workflows. Understanding the performance and efficiency of these workflows is key to productivity, knowledge and model sharing, and energy efficiency. Even though there have been efforts in studying and designing portability protocols, performance analysis of large-scale ML is still an expert-driven task, tightly locked-in to specific physical and software infrastructure. Much like in other domains, this hinders reproducibility of both results and overall workflow performance. To overcome this challenge, we propose the design of a container-based framework for reproducible performance analysis of ML workflows at scale. We validate our framework using a case-study on two different large-scale production systems running ML workflows. We show empirically that our containerized approach is portable and allows arbitrarily low-level performance evaluation when run on two different, production-based HPC clusters with hundreds of GPUs. We report our findings on widely-used open-source software stacks and datasets and offer practitioners insights into what types of analyses our framework enables. To benefit the community, we open-source our software and results.

## 1 Introduction

The rapid advancements in hardware performance in recent years have enabled the operation of machine learning (ML), and especially deep learning (DL) [34]. This field has gained immense traction and interest, and made important contributions to many other domains such as medicine [56] or physics [11]. This led to an abundance of trained models or systems [1, 8, 29, 41] to run such models. Dean et al. [15] show that articles published per year in the field grow faster than compute power per Moore's law, up to *100 articles per day* at the end of 2018. Naturally, domain scientists and practitioners need to use such models and techniques to solve problems (more efficiently), and hence need to replicate the findings and setups of large amounts of ML published work.

Reproducibility [27, 46] is the Achilles' heel in computer science in general, and is much more difficult to achieve in large-scale computer systems [19, 53]. This is because the sheer complexity of the involved physical infrastructure, interconnected through large-scale networks and many layers of software. ML scientists and practitioners not only need reproducible results, but also reproducible *performance analysis*. Understanding the performance of ML models and frameworks is key to achieving productivity, knowledge and model sharing, as well as energy efficiency. This is especially important since training has been shown to have significant environmental impact [49] for several ML models.

Although reproducible results are generally difficult to achieve, seminal work [47, 48] has been steering the community toward achieving this goal. Instead, in this paper we explore the domain of *reproducible performance analysis* in large-scale distributed ML. This is a significant and challenging problem exacerbated by two aspects. First, most of the tools and systems involved are locked-in to specific infrastructure, such as HPC clusters and supercomputers. Second, large-scale infrastructure is intrinsically variable in hardware performance [17, 37], which subsequently affects application performance. Although guidelines for reproducible performance evaluation exist [40, 52], it is unclear whether these are sufficient for ML performance evaluation.

Due to their high demand of compute power, ML and DL workloads are naturally suited for deployment in large-scale HPC clusters equipped with special hardware, such as GPUs, FPGAs, or TPUs [15]. Being deployed in HPC infrastructure, ML frameworks such as pytorch [41] or Tensorflow [1] have evolved to run through specialized, tightly-coupled MPI [45] interfaces. Although several performance evaluation frameworks for MPI applications exist, like Tau [44], Scalasca [22], or VAMPIR [31], these are insufficient to assess the performance of ML workloads on HPC clusters. This is because ML and DL frameworks have many levels of complexity and use specialized hardware for which practitioners have to understand bottlenecks as well. Special lower-level profilers like nvprof [10], or pyprof[1] are needed to gather lower level metrics. Finally, all these metrics and measurements have to be combined at arbitrary levels in the software stack to understand the performance of specific components.

Moreover, reproducible performance analysis has become even more difficult due to the novel development of complex ML workflow pipelines. These workflows tend to continually expand and include increasingly complex models,

---

[1] https://github.com/NVIDIA/PyProf

pre-processing pipelines for data augmentation [13], diverse data formats and dimensionality, or even complex simulators [4, 9]. The hardware ecosystem used for training these complex systems is evolving and becoming more diverse and heterogeneous, making reproducible performance analysis difficult. Furthermore, the low-level kernels implementing key ML primitives, on which high-level frameworks depend, are also in continuous development and contribute to the complexity of these workflows. The previously-reported artificial intelligence reproducibility crisis [27] is growing at an accelerated pace and covers the whole spectrum: from numerical reproducibility to performance reproducibility. In this work we focus on addressing the latter.

Although several steps have been taken toward achieving in-depth performance evaluation for ML workloads, these are not fully reproducible and do not support complex workflows. Building blocks for performance evaluation include visualization techniques [32], lower level performance characterization [5, 28], or benchmarking efforts [7, 30]. To enable practitioners to use such systems and benchmarks over a variety of infrastructure and with better reproducibility guarantees, in contrast, in this paper we present a framework for reproducible performance analysis of ML workloads.

Our contribution is a containerized profiling framework, called SURFBoard, that operates on all modern container-enabled large-scale HPC infrastructure. In this paper we focus on performance analysis for computer vision DL workflows. However, our work is modular and highly configurable and thus can be adapted to more general ML pipelines. As a consequence, the user can extend it using other profiling tools next to our current toolkit: pytorch [41], NVIDIA DALI, Horovod [33], OpenMPI [20], TAU [44], NVProf [10]. Using this toolset, we show that we can perform performance analysis at arbitrary levels of the software stack. Users can easily answer questions such as *what is the training time scalability?*, *what parameters affect batch duration?*, or *what are the most time-consuming CUDA kernels per batch?*.

In helping practitioners with performing reproducible performance analysis on complex ML/DL workflows, we show that SURFBoard is able to capture complex performance behavior on two different production-based GPU-enabled clusters. Our experiments focus on implementing typical, real-world analyses that practitioners use to search for bottlenecks and inefficiencies in their training workflows. To enable reproducible performance evaluation in ML workflows on large-scale HPC infrastructure, our contributions are:

1. We present the design and implementation of SURF-Board, a containerized profiling framework for ML workloads. To benefit the community, we open source[2] our work as well as the visualization notebooks and collected performance datasets (Section 2).

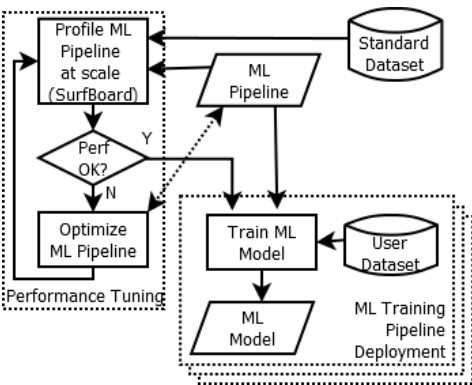

Figure 1: SURFBoard in the typical ML Pipeline life-cycle.

2. To validate our work, we present a case-study of performance analysis for DL workloads on two large-scale GPU-enabled clusters. We provide an in-depth experiment design showcasing the features of our profiling framework when running typical performance analysis that practitioners do when benchmarking their DL workflow. (Sections 3- 4).

3. We present an in-depth performance analysis on the two clusters using real-world open-source frameworks, workloads, and datasets. We show the portability and reproducibility of our results, discuss the main findings of our experiments, and show how practitioners can use SURFBoard to identify bottlenecks and performance issues (Section 5).

## 2  SURFBoard: Containerized Profiling Workflow Design

In this section we describe in detail the design and implementation of our containerized profiling framework called SURFBoard. Our work is able to integrate with any ML or DL framework running on high-end HPC infrastructure. We show how we integrate our profiling workflow with state-of-the-art ML software stacks, such as pytorch, openMPI, DALI,[3] or Horovod [43]. We show how ML workflows can be reproduced and ported to large-scale infrastructures that differ in software and hardware through containerization, and how users can perform parameter sweeps over important parameter spaces.

Figure 1 illustrates the typical life-cycle of a ML pipeline. The pipeline consists of training scripts and possibly container definitions for training a specific ML model, such as ResNet-50. Starting from the initial pipeline, profiling is performed at scale, using some representative dataset, and the pipeline is optimized until performance meets some user-defined criteria. SURFBoard helps automate this part of the life-cycle.

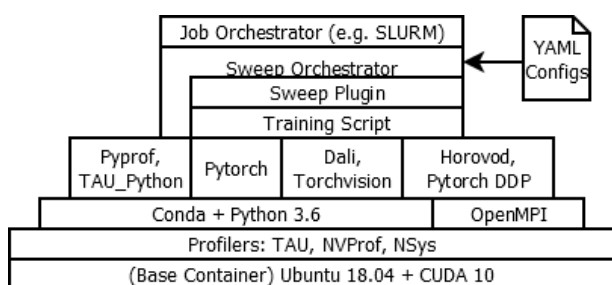

Figure 2: Software stack for profiling ML workflows.

Subsequently, the optimized ML pipeline can be deployed to train ML models many times, on various datasets.

In most cases, profiling starts with the user requesting a job orchestrator, e.g. SLURM, for a node allocation and subsequent execution of the training algorithms needed to run. Developers and practitioners may want to perform a parameter sweep over the possible parameter space: input dataset, number of I/O workers, number of GPUs per node, batch size, gradient precisions etc. To enable this, we implemented a *Sweep Orchestrator* with a specified set of YAML configuration files. The orchestrator utilizes the application-specific Sweep Plugin to generate a set of command line invocations of the training script inside a containerized environment, using MPI for inter-process communication.

To achieve portability and performance reproducibility over an extensive set of large-scale infrastructure, different in both hardware and software deployments, we created a containerized environment that can run state-of-the-art ML training stacks on most (high-end) HPC infrastructures. One of the most challenging issues in performance reproducibility is achieving similar setups on different infrastructure. We solve this by creating our containerized approach to ML training performance analysis.

Figure 2 illustrates the software stack involved in executing profiling experiments with our workflow. The stack is roughly divided into two sections consisting of code running in a host environment and code executed inside a Docker or Singularity container. The container is configured to provide communication and profiling infrastructure in addition to the training capability of Pytorch and associated libraries for neural network training. The specific capabilities of the container are:

**1. Training** with Pytorch and optionally using the Nvidia Data Loading Library, DALI. Compared to the Pytorch built-in dataloader library, Torchvision, DALI offers multiple advantages: input from folders of images or Tensorflow Records (TFRecords), various levels of GPU offloading of the preprocessing, and advanced profiling. We provide a custom-built version of DALI with NVTX annotations enabled, which allows GPU profilers to inspect the details of preprocessing-related computation. It is important to note here that our entire pipeline is configurable, and the user can add different

training frameworks, like Tensorflow [1].

**2. Communication** over OpenMPI, which can be leveraged from Pytorch either through the Pytorch built-in native multi-node training module, Pytorch DDP, or through Horovod. We include Horovod specifically for its capability for explicit gradient quantization, i.e. casting the Pytorch-computed FP32 gradients to FP16 before invoking MPI to perform all-reduce for data-parallel training. We configure Horovod to perform all its operations with MPI even when transferring from GPU memory, which enables MPI profilers to inspect the traffic.

**3. Profiling** at multiple levels, targeting compute and communication. The MPI-profiler leveraged in this work is TAU [44], used for detailed communication profiling. NVProf and NSys are leveraged for detailed compute kernel profiling on the GPU, and Pyprof for linking GPU profiling data to the Python execution graph and neural network model.

We stress that the tools used in our work are mere examples of what practitioners can achieve with a containerized framework for reproducible performance evaluation. In fact, all three layers of our containerized design are highly configurable, and other types of tools could be integrated. For example, one can gather CPU performance counters using PAPI [55], or use VAMPIR [31] instead of TAU.

## 2.1 Sweep orchestrator

We denote a profiling sweep as the set of profiling experiments which traverses all possible application parameter combinations. As this set can be relatively large and varies between neural networks, we provide experiment orchestration infrastructure based on Hydra[4] designed to enable automatic generation of command line arguments for a profiling sweep. We note that each experiment command follows the structure: *mpirun <MPI args> singularity exec <singularity args> <profiler args> main.py <application args>*

The orchestrator defines a base Python class Experiment, initialized with the complete set of possible experiment parameters, as well as necessary abstract functions for parameter combination filtering and command string generation. For non-numeric parameters, the orchestrator defines Enums which constrain the parameter values. The possible application parameters are defined in Table 1, and divided into four classes corresponding to the four types of CLI arguments in the command structure:

**1. Scale and infrastructure parameters**, which manifest in the MPI arguments, include the number of nodes, GPUs per node, CPUs per process, and the networking fabric utilized for communication (Infiniband or Ethernet)

**2. Container parameters** include, for example, the singularity container itself (SIF file), as well as the locations of training data and TFRecord index, if needed. These are

---

[4]https://github.com/facebookresearch/hydra

| | **Possible Values in YAML Config** | **High-level Description** |
|---|---|---|
| Ranks | Numeric | Total number of MPI Processes executed |
| GPU per Node | Numeric | Processes allowed per-node, each of which maps to one GPU |
| Profile Level | None, tau_exec, tau_python, tau_python_cuda, nsys, nvprof | No profiling
Use TAU for mpi, python and GPU profiling respectively
Use NVidia tools for GPU profiling |
| Network Backend | ib, eth | Data transfer fabric (InfiniBand or Ethernet) |
| Repetitions | Numeric | Number of runs of each experiment configuration |
| Gradient Precision | fp16, fp32 | Precision of the communicated gradients. |
| Compute Precision | fp16, fp32, mixed | Precision of Compute |
| Batch size per GPU | Numeric | Size of per-GPU batches |
| Data Loader | pytorch, dali-gpu, dali-cpu-to-gpu | Use Torchvision dataloader
Perform all preprocessing on GPU
Perform all preprocessing on CPU, then move data to GPU |
| Data Format | folder, tfrecord | Use compressed images in specified folder
Use pre-packaged Tensorflow Records |
| Workers | Numeric | Number of preprocessing worker threads |
| Communication Backend | Horovod [43] | Gradient synchronization framework |

Table 1: Typical parameters that DL practitioners consider in their performance evaluations and their high-level description.

mounted into the Singularity container at pre-defined mount points.

**3. Profiler parameters** define the profiler to be utilized as well as any parametrization of the profiler. Profiler options are TAU, TAU-Python (with optional CUDA support), or NVProf/NSys CUDA profilers.

**4. Training parameters** configure the NN training flow itself: input data format (folder of images or TF records), data loader (torchvision, DALI CPU, DALI GPU), execution precision – fp32, fp16, or mixed, gradient precision between fp32 and fp16, and distribution backend (Horovod).

## 2.2 Implementing New Training Workflows

We offer the user the possibility to define new types of experiments to enable novel training workflows while retaining the capability of achieving portable and reproducible performance evaluation. To plug a new training flow into the Orchestrator, the user must define a new class inheriting Experiment and implement the cmd() method which produces a command from the relevant parameters as well as the is_legal() method, which filters out parameter combinations that the target training flow cannot implement.

At run-time, the orchestrator leverages Hydra to assemble a sweep configuration from YAML files as follows: a main configuration file defines which parameters have constant values across all runs in a sweep and which cycle through multiple values. The possible values of non-constant parame-

ters are defined in a second YAML configuration file. Hydra assembles the information about parameter values, calculates possible parameter value combinations as the Cartesian product of possible values for each parameter, checks the legality of the parameter combination using is_legal(), and executes the command produced by cmd() from the specified parameter values. To define a new sweep, the user only needs to define the two YAML configuration files.

## 2.3 Open Source Commitment

Our work is implemented in python. Development took approximately 6 person-months, most of which was spent debugging the various components of the deep software stack illustrated in Figure 2, setting up the NVTX profiling infrastructure, and to ensure portability of containers and experiment orchestrator between various systems.

The experimental data gathering and visualization took another 6 person-months. We release both the source code of our reproducible performance analysis framework, as well as all the performance data and visualization scripts. Since the start of the project in 2019, approximately 50,000 core hours were spent for debugging and initial framework calibration, while 300,000 core hours have been utilised on the Cartesius cluster and 50,000 core hours have been used on the LISA cluster (see Section 4 for an in-depth description of these clusters) for running profiling experiments. In the following sections of this paper we present our validation of SURFBoard

using a case-study of large-scale training experiments on two production HPC infrastructures.

## 3 Case Study: Experiment Design

In this section we describe in detail the design of the experiment we perform to validate SURFBoard. We seek to empirically show that our performance analysis framework adheres to reproducibility standards and is able to help ML practitioners answer valuable questions about the performance of ML training workflows. We describe the high-level goals of our experiment, which are typical questions a ML practitioner would ask when assessing the performance of a ML workflow. We focus on the methodology of performing the performance analysis of the high-level goals, and we describe in detail the DL model used in our study.

### 3.1 Case Study High-level Goals

The goal of this profiling exercise is to evaluate compute and communication efficiency for data-parallel distributed training of ResNet50 on SURFSara infrastructure, and quantify the contribution of each training pipeline stage (batch preprocessing, training, communication) to the total runtime, under various configurations of each stage. Furthermore, we wish to construct a performance model enabling performance extrapolation. We separate this goal into three large sub-objectives:

**1. Scalability.** We aim to determine the effect of various training configurations on the scalability of training up to the maximum sizes permitted by the hardware. We measure the scaling efficiency itself but also how each configuration option affect the scaling efficiency at each scale.

**2. Computation Efficiency.** We measure each stage in the training process: forward, backward, and model update, as well as the total batch duration, and calculate the overall compute efficiency of the GPU, as well as the overall memory bandwidth efficiency achieved by the GPU.

**3. Preprocessing Computation.** We compare CPU and GPU preprocessing via total application run-time at various scales in order to determine the effect of the number of preprocessing workers and preprocessing offload on run-time at various scales.

**4. Adhering to Reproducibility Standards.** We seek to determine whether our performance analysis framework is able to run and achieve significant results on multiple types of infrastructure. We compare the results of our framework on two different large-scale production HPC systems. The practical details of these systems are detailed in Section 4.

### 3.2 Performance Analysis Method

**1. Important Parameters.** When performing DL performance analysis, practitioners usually focus on several important parameters. In this study, we consider the impact of the following parameters: *gradient and compute precision*, *size of the batches*, *data loader*, and *number of workers*. These parameters are described and explained in Table 1.

**2. Scalability.** To study the scalability we measure both the duration of the experiments at various scales and the scaling efficiency (SE) for $N$ GPUs. The duration of one experiment is measured using the data from TAU as the duration of the *.tau application*. Since the application is running on several GPUs, the maximum duration over all GPUs is used as the final measure. The scaling efficiency is measured as the ratio between the experiment duration using a baseline number of GPUs (e.g., one GPU) over the duration of the same experiment using $N$ GPUs:

$$SE_N = \frac{t_{baseline}}{t_N}$$

**3. Efficiency.** To study the computational efficiency, we perform deeper analysis by measuring the batch duration and the duration of the three stages of each training iteration: *forward pass, backward pass, parameters update*. During the forward pass, the DNN makes a prediction of the labels associated with each image in the input batch, and an error is calculated by comparing the known correct labels with the predicted ones. In the backward pass, the error gradients are calculated and propagated through each network layer. Finally, the gradients are utilized to update each network parameter to minimize the error.

We use NVProf along with NVTX annotations to delimit the previous stages. We also calculate the overall compute efficiency of the GPU using the CUDA-kernel-level data from Pyprof. The number of floating points operation per second is measured as the ratio of the sum of the FLOPs over the sum of the duration of each kernel. To obtain the compute efficiency of the GPU, we divide the measured value by the theoretical value of the given GPU. Similarly, the memory bandwidth efficiency is computed as the ratio of the measured bandwidth (ratio of total amount of bytes in and out of the GPU over the sum of the duration of each kernel) of the GPU over its theoretical bandwidth.

**4. Sensitivity Analysis.** We study the impact of the configuration parameters over the scaling efficiency of the application by using Taguchi Methods [50]. The goal of such methods is to reduce the number of experiments needed to be performed in order to determine which factor(s) impacts a predetermined target variable the most. We do not use the method to design our experiment but only to evaluate the parameter importance given our experimental results. For that, for a given experiment, we use the signal to noise ratio (SN) defined as follows for the Taguchi Methods:

$$SN = -10\log\left(\frac{1}{N}\sum_{i=1}^{N}\frac{1}{y_i^2}\right)$$

$N$: Number of repetitions of the given experiment;
$y_i$: Target variable value for repetition $i$ of the experiment.

## 3.3 DL Model Used in the Study

We perform our experiments on a state-of-the-art, industry-standard model and dataset: ResNet50 v1.5. We use training scripts implemented by Nvidia as part of their state-of-the-art reference examples in Pytorch [39]. The model and training scripts are configured for image classification with the ImageNet dataset.

Compared to the original definition [23], ResNet50 v1.5 has stride = 2 in the 3x3 convolutions, rather than in the 1x1 convolutions, in the bottleneck blocks that require downsampling. This comes at a small increase in terms of computational cost, but is beneficial in terms of accuracy. This modification was first introduced in a Lua Torch re-implementation of ResNet from Facebook [18], and has since been widely adopted. A more detailed overview of ResNet variants can be found in [24], where ResNet 1.5 is referred to as ResNet-B.

Nvidia's scripts provide an implementation that is highly tuned both in terms of hyper-parameters and final accuracy, as well as training-time performance. As such, it is more representative of the current state-of-the-art compared to Pytorch's reference ImageNet training implementation [42].

With respect to performance, Nvidia's implementation tightly integrates with the DALI library for data loading and pre-processing. DALI has multiple advantages over Pytorch's own dataloaders: it provides support for reading input data stored in the Tensorflow TFRecord format, which we leverage as part of our setup; it provides partially GPU-accelerated JPEG decoding and end-to-end GPU-accelerated preprocessing for the ImageNet dataset. With respect to accuracy, they implement all the strategies described in [24], that together contribute to pushing the top-1 ImageNet accuracy to around 78.4%.

It is important to note that DL model described above is used as an example for validation study and to showcase capabilities of the framework and type of the analysis integrated within the framework.The framework is designed with principle of extensibility and will require some effort from ML practitioners / developers to integrate their model within the framework. The focus of this research is to highlight the approach and the methodology rather than the results themselves on a specific DL model.

## 4 Experiment Setup

For our experiment we target two production-grade distributed infrastructures The two large-scale HPC clusters we run our experiments on are LISA[5] and Cartesius,[6] specifically their GPU islands. Note that both hardware and software stacks of the two systems are highly different. We show empirically that the containerized performance analysis workflow we propose is portable and produces reproducible results that

---

| Software / Library | Version |
| --- | --- |
| PyTorch | 1.2.0 |
| Python | 3.6 |
| CUDA | 10.0 |
| DALI | 0.18.0 |
| TAU | 2.28.1 |
| PyProf | 3.6.0 |
| CuDNN | 7 |
| OS | Ubuntu 18.04 |

Table 2: Software versions for the container environment.

| | Cartesius | LISA |
| --- | --- | --- |
| Nodes | 8, 16, 32, 48 | 1, 4, 8 |
| GPU per Node | 2 | 4 |
| Gradient Precision | fp16, fp32 | fp16, fp32 |
| Compute Precision | fp32 | fp32 |
| Batch size per GPU | 32, 64 | 32, 64 |
| Data Loader | dali-gpu, dali-cpu-to-gpu | dali-gpu, dali-cpu-to-gpu |
| Workers | 2, 8 | 2, 4 |

Table 3: Parameters considered for the experiment on both Cartesius and LISA systems.

could be compared between the two. Note that these two types of production-ready clusters are comparable to what ML and DL practitioners use in practice to deploy training workflows. We have not chosen any more clusters in our results because we would like to focus more on showcasing methodology and approach using SURFboard than the results themselves. We would like to encourage HPC community to expand and validate this approach on more infrastructures.

### 4.1 Cartesius Hardware Specification

The Cartesius GPU island consists of 66 Bullx B515 processing nodes. Each node is equipped with a 16-core E5-2450 v2 Intel CPU (Ivy Bridge microarchitecture), operating at 2.5 GHz, and 96 GB of memory. Each node is also equipped with two K40m GPUs, and two Mellanox Connect-X3 Infiniband adapters, with a maximum throughput of 56 Gbps each. For our experiments we utilized up to 48 nodes. Cartesius is maintained with RedHat 4.8.5-39, Linux version 3.10.0-1127.8.2.e17.x86_64. We have used CUDA enabled OpenMPI/3.1.2 for transferring data buffers directly between GPUs using Infiniband network.

### 4.2 LISA Hardware Specification

The LISA cluster consists of 25 GPU-accelerated nodes, each equipped with Intel Xeon Bronze 3104 CPUs (12-core, 1.7 GHz), 256 GB of memory, and four GPU accelerators, either

NVIDIA GeForce 1080Ti or NVIDIA Titan V GPUs. The nodes are connected through 40Gbps Ethernet. LISA is maintained with Debian GNU version 10 (buster). We have used OpenMPI/3.1.4 for multinode scaling experiments.

## 4.3 Software Environment inside the Container

Table 2 outlines the software environment inside the container for both LISA and Cartesius. This is one of the main advantages of using containers. It enables the use of the same software environment on both HPC clusters, enabling reproducibility over many types of software and hardware infrastructure.

## 4.4 Achieving Empirical Reproducibility

We profile the communication of the application with TAU both on Cartesius and LISA using combinations of the parameters presented in Table 3. In order to gather a statistically valid sample, we conduct at least 10 experiments for each combination of parameters, each of them being run for a total of 50 batches. Since our analysis is performed by gathering metrics at batch level, we ensured that in total 500 batches per experiment achieves statistical significance and is in check with current reproducibility standards [37, 40, 52]. In our figures, we present the median of a given metric over the 10 experimental runs along with the 95% confidence interval for the communication data. Additionally, we conduct GPU profiling on Cartesius gathering 10 experiments for each combination of parameters in Table 3 for a total of 25 batches using NVProf, and 25 batches using NVProf and enabling kernel profiling via Pyprof.

## 5 Results and Visualization

In this section, we showcase results and visualizations of data that can be produced using the framework presented in this paper. We present the data from higher (experiment duration and communication) to lower (GPU efficiencies and CUDA kernels) levels of the software stack. The experiments we performed are typical analyses performed by DL practitioners and the conclusions we draw can help practitioners build training infrastructure that is suitable to their workloads, identify bottlenecks, and identify what are important parameters in their setups. Moreover, this kind of analysis shows that SURFBoard is useful in helping practitioners analyze the performance of their DL workflows in a reproducible manner, across multipe types of infrastructure.

**Lessons Learned.** The main lessons learned from analyzing the empirical experiments performed in our study are the following:

1. We confirm that Pytorch, when coupled with Horovod, achieves a good scalability ($> 90\%$) for ResNet-50-like workloads, see Figures 3, 4.

2. On infrastructure like LISA and Cartesius, where resources are not shared, there is not much overall performance variability, especially on the MPI collective operations, see (for example, whiskers in) Figures 5, 6.

3. The computation throughput of ResNet-50-like workloads is neither memory-bound, nor compute-bound. The bottlenecks lie in waiting for remote data from other GPUs to be transferred by the DL framework, see Section 5.3.

4. On some types of machines, the CPU-to-GPU ratio is important, as the GPUs need to be fed data quickly enough to achieve good performance. Our framework can be used by system designers to detect bottlenecks like these and areas of improvement. Users could perform similar steps of analysis using their own workloads to determine an appropriate ratio of CPUs to GPUs, see Section 5.4.

5. The number of preprocessing workers has more importance than the batch size on LISA. However, the opposite behavior is true in architecture like Cartesius. Practitioners should perform similar types of analyses to decide what parameters are most important in their workloads and how the performance could be improved by using this knowledge, see Section 5.4.

6. SURFBoard is able to be deployed on two different large-scale HPC infrastructures. It can further help practitioners identify behavioral differences on large-scale infrastructures and what deployment parameters cause these.

## 5.1 Scalability

**Execution Time.** We measure the duration of the experiments as the total runtime of the TAU application on a GPU involved in the computation. Experiment durations for both Cartesius and LISA are presented in Figures 3 and 4 respectively. For both systems, the duration scales linearly with the number of GPUs. This is likely due to the communication overhead being increasingly more significant for higher numbers of GPUs used. On Cartesius, experiments using fp32 gradients all take longer than the ones with fp16 gradient. This behavior is the opposite on LISA.

Since gradient casting requires additional CPU cycles, and the GPU to CPU thoughput ratio on LISA is higher than on Cartesius, we hypothesize that this difference in behaviour on the two systems is caused by a CPU-induced bottleneck on LISA. These results illustrate the utility of our framework to system designers, e.g. our results would indicate provisioning more CPUs to LISA as a relatively inexpensive way to increase DL performance. Alternatively, gradient casting

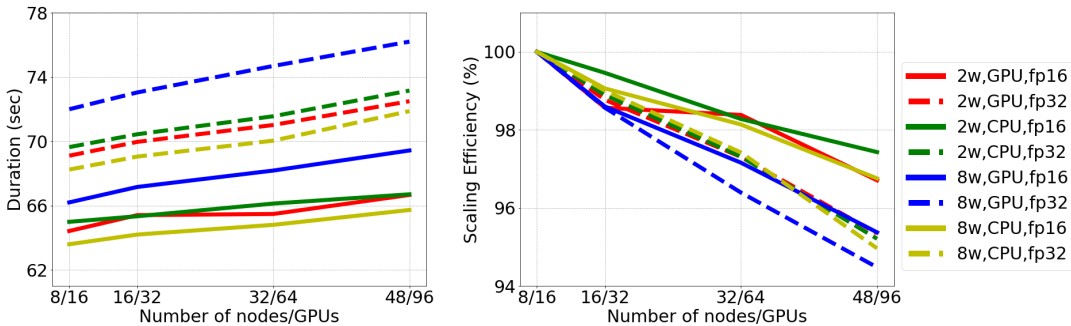

Figure 3: Duration and scaling efficiencies of 50-batch experiments on Cartesius with different configurations. Experiments depicted uses 32 images per batch. Legend reads as follows: *<number of workers>*w,*<preprocessing>*,*<gradient precision>*. Note: the vertical axis does not start at 0 for better visibility.

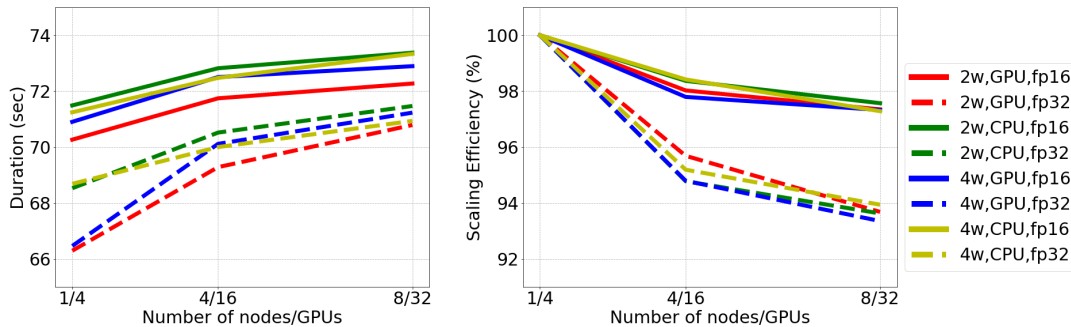

Figure 4: Duration and scaling efficiencies of 50-batch experiments on LISA with different configurations. Experiments depicted uses 32 images per batch. Legend reads as follows: *<number of workers>*w,*<preprocessing>*,*<gradient precision>*. Note: the vertical axis does not start at 0 for better visibility.

should be offloaded to the GPU or some other accelerator to relieve the bottleneck.

**Scaling efficiency.** We compute and present the scaling efficiencies of the application for both Cartesius and LISA in Figures 3 and 4, respectively. The efficiencies are computed as the ratio of the duration of the given experiment over the baseline experiment. For the experiment on Cartesius, the baseline was taken as the expriment using 8 nodes (16 GPUs) whereas the baseline on LISA was taken as the experiment using 1 node (4 GPUs). We used different baselines for computing scalability to show the flexibility of our framework, and to model practitioners' behavior when scaling DL computations: for large-scale training, using few resources it too time-consuming and early scale-out is needed. Our experiments also show that using fp32 gradients results in lower scaling efficiency while using CPU preprocessing gives a larger one on both systems.

## 5.2 Communication

Our framework uses TAU to profile communication through several MPI call metrics. Figure 6 and 5 presents the sum of the duration of MPI_Allreduce, MPI_Bcast and MPI_Gather across all GPUs. All of these are considered extremely important for DL performance by practitioners. It is also possible to gather other metrics such as number of MPI calls, and total volumes of messages sent across GPUs, see Figures 7, 8, 9 and 10. MPI_Allreduce is used during the model update phase to share the gradients of each weights of the neural network. Resnet50 has approximately 23 million parameters. The experiment runs on each GPU for 50 batches and the gradients are stored using 2 or 4 bytes (half or full-precision floating point precision format). As a consequence, we expect the total volume of message exchanged for MPI_Allreduce to be:

$$Volume = N_{weights} \times grad\_prec \times N_{batches} \times N_{GPU},$$

where $grad\_prec$ is the number of bytes used to store each gradient (4 bytes for fp32 gradients); $N_{batches}$ is the number of batches (50); $N_{GPU}$ is the number of GPUs. The total volume of messages exchanged for MPI_Allreduce presented on Figure 9 and 10 is in line with the expected volume. Using similar types of analyses, practitioners can identify bottlenecks or ill-behavior at the MPI collective operation and networking layer when performing large-scale training.

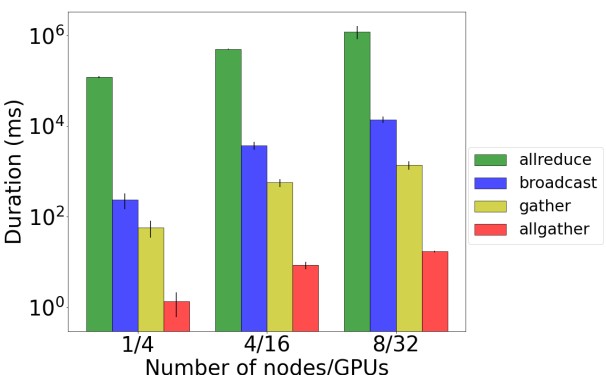

Figure 5: MPI calls durations on LISA for 2 workers, CPU preprocessing, 32 images per batch, gradient fp32. Note: bars are the median values over 10 runs, whiskers 95% confidence interval, vertical axis is logarithmic, lower is better.

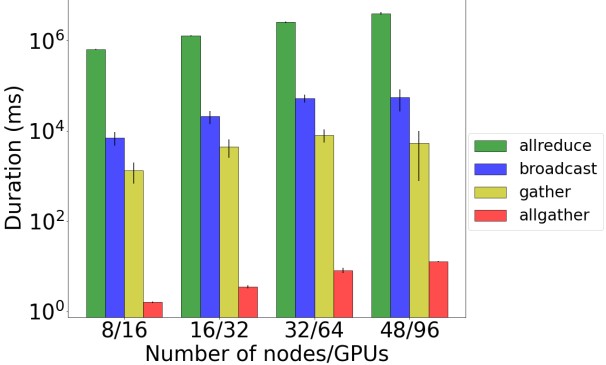

Figure 6: MPI calls durations on Cartesius for 2 workers, CPU preprocessing, 32 images per batch, gradient fp32. Note: bars are the median values over 10 runs, whiskers 95% confidence interval, vertical axis is logarithmic, lower is better.

### 5.3 Compute and Memory Efficiency

**Batch duration.** Our framework combines NVProf along with NVTX annotations to delimit the training stages (forward, backward, update) and obtain more details about the training. The batch duration and training-stages duration can be visualized in Figure 11. We observe that the duration of the batches scales with the number of nodes/GPUs. In particular, the backward phase of the training, which includes gradient synchronization over InfiniBand, takes longer for larger number of GPUs whereas the forward and update phases stay constant. We also note the increased variability in batch duration at larger scales, caused by corresponding variability in the time required for gradient synchronization over the network fabric and the effect of uncorrelated performance jitter between the GPU workers, which can have a variety of causes - OS scheduling, resource contention, garbage collection. Practitioners can make use of this type of analysis to decide which

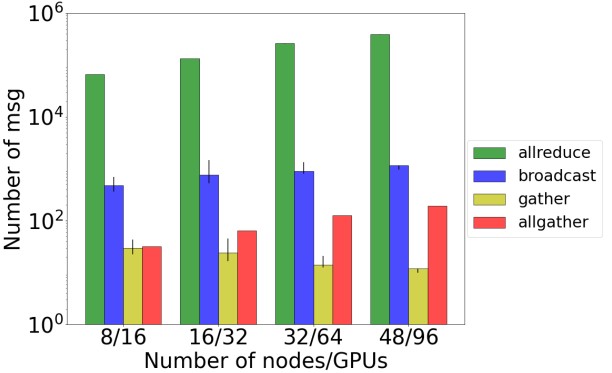

Figure 7: Number of MPI messages exchanged across all GPUs on Cartesius for 2 workers, CPU preprocessing, 32 images per batch, gradient fp32. Note: bars are the median values over 10 runs, whiskers 95% confidence interval, vertical axis is logarithmic, lower is better.

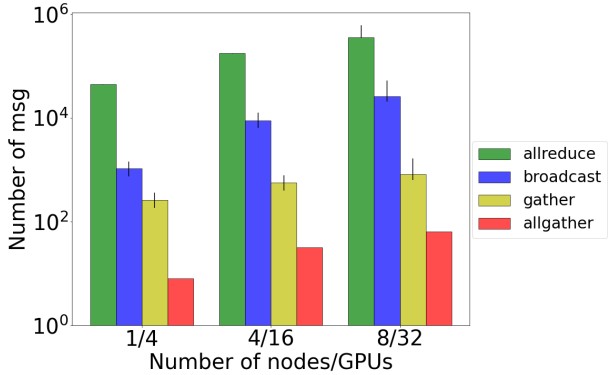

Figure 8: Number of MPI messages exchanged across all GPUs on LISA for 2 workers, CPU preprocessing, 32 images per batch, gradient fp32. Note: bars are the median values over 10 runs, whiskers 95% confidence interval, vertical axis is logarithmic, lower is better.

parts of the per-batch computation are bottlenecks or variable in performance.

**GPU performance metrics.** Using Pyprof, we measure kernel-level data and compute the utilized memory bandwidth and utilized compute capacity of the GPUs on Cartesius. We present the results in Table 4 along with the efficiencies relative to the theoretical performance of the specific GPU model we run experiments on. NVIDIA Tesla K40m has a peak memory bandwidth of 288.4 GB/s and a theoretical compute performance using fp32 float of 5.046 TFLOP/s. Table 4 shows that both memory and compute efficiency are low (below 18%). This is due to the mismatch in model (implementation) and the hardware we tested on. Given the low computed memory bandwidth and compute efficiencies, each below 20%, it seems that the application is neither memory nor compute bound. Low GPU utilization (below 50%) is

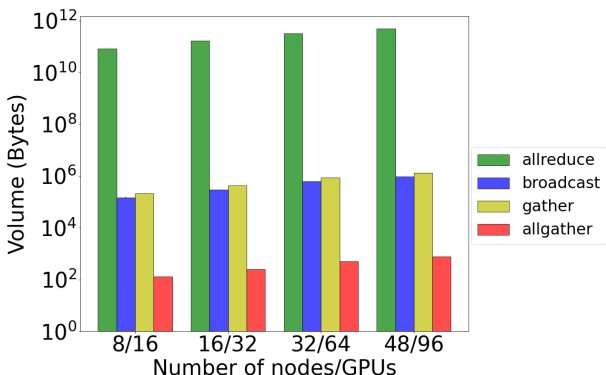

Figure 9: Volume in bytes of MPI messages exchanged across all GPUs on Cartesius for 2 workers, CPU preprocessing, 32 images per batch, gradient fp32. Note: bars are the median values over 10 runs, whiskers 95% confidence interval, vertical axis is logarithmic, lower is better.

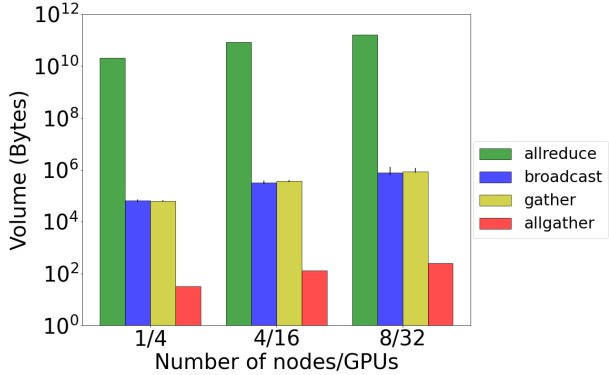

Figure 10: Volume in bytes of MPI messages exchanged across all GPUs on LISA for 2 workers, CPU preprocessing, 32 images per batch, gradient fp32. Note: bars are the median values over 10 runs, whiskers 95% confidence interval, vertical axis is logarithmic, lower is better.

expected for deep learning frameworks, as noted in [12, 57]. Using this kind of analysis, practitioners can identify which parts of the GPU implementation represent bottlenecks.

**CUDA kernels.** Pyprof also allows to retrieve a detailed kernel summary containing, among other metrics, the time elapsed, number of bytes in and out of the GPU, number of floating points operations performed during the execution of a given kernel. Table 5 presents the duration of the 10 most used kernels during a single batch. According to [16], the top four kernels in Table 5 correspond to backwards pass convolution, forward pass convolution, fully connected layer (forward and backward), and element-wise addition respectively. These results are in accordance with expectations given the structure of the ResNet50 DNN. Using this type of analysis, practitioners can visualize what GPU kernels take the most GPU computation time and identify potential bottlenecks.

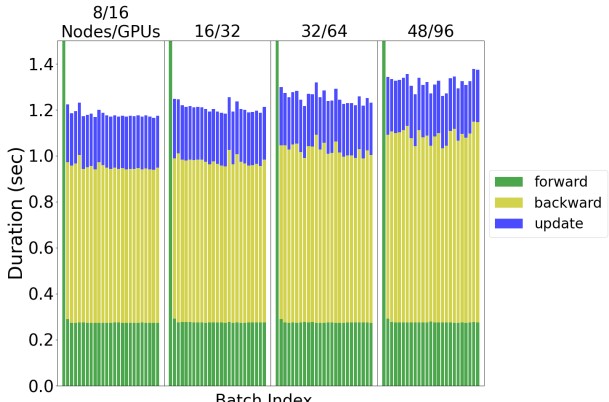

Figure 11: Scaling of training stages duration on Cartesius. Configuration presented: 2 workers, dali-cpu-to-gpu, 32 images per batch, gradients fp32, compute fp32. lower is better.

| GB/s | Bandwidth efficiency | TFLOPs/s | Compute efficiency |
|------|---------------------|----------|--------------------|
| 51.16 | 16.23 % | 0.6964 | 17.77 % |

Table 4: NVIDIA Tesla K40m GPU performance measures on Cartesius for 8 nodes, 2 workers, CPU preprocessing, 32 images per batch, gradient fp32. Both efficiencies are computed with respect to the theoretical performance of the GPU.

## 5.4  Parameter Sensitivity

To get a more detailed understanding on the effect of each parameter we considered in this work, we study the impact of the number of GPUs, number of workers, type of dataloader, size of the batches, and gradient precision on the scaling efficiency using Taguchi Methods, as described in Section 3.1. To do so, we compute the range of SN ratio for each of the parameters based on the set of experiment configuration performed. Since the goal is to determine the ranking of the parameters for each system, we normalize the range of SN ratio and present the order of importance on Figure 12a for Cartesius and Figure 12b for LISA. The larger the range of the SN ratio is, the more impactful a parameter is. Disregarding the number of GPUs which is obviously the most impactful parameter of the scaling efficiency, on both systems, the gradient precision is the second-most important parameter. Because the gradients are sent between nodes/GPUs, smaller precision gradient results in lower overall volume of data exchanged via MPI and therefore a better scalability of the application. The least impactful parameter on both system is the dataloader. Interestingly, batch size and number of workers switch places in their effect on scaling efficiency on the two systems under analysis (note the different coloring scheme for *number of workers* and *batch size* in Figures 12a and 12b). Conceptually, batch size should be the more important parameter to scalability, as it directly correlates to

| Kernel name | Time (s) | % total |
|---|---|---|
| cudnn::detail::wgrad_alg0_engine | 0.2597 | 15.37 |
| cudnn::detail::implicit_convolve_sgemm | 0.2395 | 14.17 |
| sgemm_sm35_ldg_nt_64x16x64x16x16 | 0.1679 | 9.935 |
| elementwise_kernel | 0.1522 | 9.009 |
| sgemm_largek_lds64 | 0.1332 | 7.882 |
| cudnn::detail::dgrad_engine | 0.1181 | 6.989 |
| sgemm_sm35_ldg_nn_64x16x64x16x16 | 0.1023 | 6.055 |
| cudnn::detail::dgrad_alg1_engine | 0.0933 | 5.521 |
| cudnn::detail::bn_bw_1C11_kernel_new | 0.0913 | 5.402 |
| cudnn::detail::bn_fw_tr_1C11_kernel_NCHW | 0.0415 | 2.457 |

Table 5: Duration and percent of total runtime of the 10 most used kernels during a single training batch on Cartesius for 2 workers, CPU preprocessing, 32 images per batch, gradient fp16.

the time spent in computing and decreases the relative importance of communication to the total application time. This expectation is confirmed on Cartesius. On LISA, the relative under-provisioning of CPU compute to GPU compute may cause the difference in relative importance of the number of preprocessing workers to scalability, although it must be noted that the absolute differences between configurations in this respect are small (see Figure 4). Using this type of analysis, practitioners can perform in-depth analysis on what kind of parameters are most important for their DL training workflows and decide on which type of other analysis to zoom-in to identify possible bottlenecks and places of improvement.

## 6  Limitations

We discuss limitations and threats to validity of our work. The scope and goal of this work is to provide practitioners with a framework for the reproducible performance analysis of ML workloads, and not in presenting an in-depth performance analysis and tuning of a given workload. Instead, we provide a case-study on a workload and two different clusters, showcasing the reproducibility of our work and the differences in performance obtained in the two clusters. SURFBoard is extensible and can be tuned to accept novel workloads, ML frameworks and container orchestration tools. To this end, we identify the following limitations of our work.

**1. Single Model** We only validated our framework on the de-facto computer vision workload and dataset. This is a widely-used workload in the ML community and we chose this because the results we gathered can be compared by practitioners against their own results. However, adding other models and datasets is feasible. We are working toward adding new models and open-sourcing our workflows and results.

**2. PyTorch** Our case-study and results are obtained only on PyTorch, which is widely used by practitioners. However,

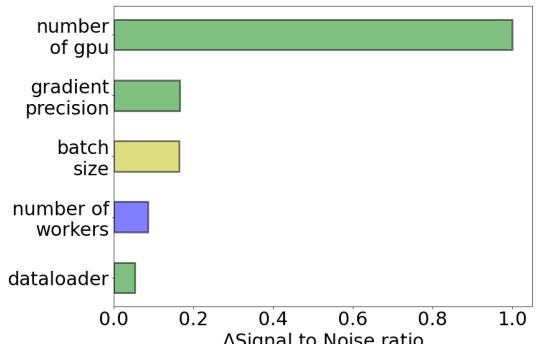

(a) Cartesius.

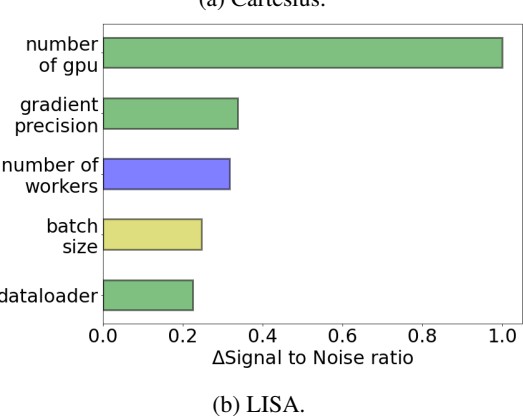

(b) LISA.

Figure 12: Ranking of experimental parameters from most to least important in order to maximize the scaling efficiency of the application on both systems. Note that the most important is the ordering of the parameters here, not their magnitudes and that the two figures are not comparable in absolute values.

swapping PyTorch for Tensorflow in our workflows is only an implementation effort. All other steps and components can be re-used from our PyTorch proof-of-concept.

**3. Difficulty of Extensibility** We have built SURFBoard with extensibility in mind, allowing each component to be replaced by others. However, we have not studied in depth, using independent programmers and practitioners how easy it is to achieve extensibility. We plan to study this in the future by performing user studies that can help us improve our framework.

## 7  Related Work

In this section we discuss work related to our approach. We identify four main categories of related work: (i) performance reproducibility in large-scale systems; (ii) ML performance analysis; (iii) ML workflows and orchestration; and (iv) ML/DL benchmarks. We discuss each in detail and contrast with our own approach.

**Performance reproducibility in large-scale systems.** Performance reproducibility in large-scale systems is an elu-

sive task. There are no community-wide agreed methodologies to achieve it, although several authors have addressed the topic. In HPC, Hoefler and Belli [26] propose a set of 12 principles to achieve credible performance evaluation. In cloud computing, which is much more variable than HPC environments [37], Papadopoulos et al. [40] propose a set of methodological principles to achieve reproducible performance evaluation. However, these seem insufficient as multiple types of variability are identified by Uta et al. [52], which highly affect reproducibility of results. In our work, we leverage all the findings of such types of work and perform sanity checks on our results. We also adhere to reproducible methods for achieving performance results, like performing many repeated experiments (during different days), computing nonparametric confidence intervals for medians, and analyzing variability.

**ML performance analysis.** Due to the inherent demands in computational requirements, ML performance analysis is currently of utmost importance. As outlined by Amodei and Hernandez [3], the amount of compute used when training the largest AI models increased exponentially with a 3.4-month doubling time, far outpacing Moore's law, resulting in a $300,000x$ increase between 2012 and 2018. Furthermore, Hernandez and Brown [25] estimate that the algorithmic efficiency also improved by a factor of $25x$ in the same period, leading to 7.5 million times increase in the effective training compute available to the largest AI experiments. Dakkak et al. [14] propose the MLModelScope toolkit that includes performance analysis along with model evaluation, in a reproducible, containerized fashion. However, the toolkit is concentrated on non-distributed training only. Modern ML models are trained in a distributed fashion, using a variety of communication interconnects (e.g. NVLink, Infiniband, OmniPath, Ethernet, PCIe), and employing different parallelization strategies. Awan et al. [6] aim to measure these characteristics and propose improvements to communication patterns, with visualization tools for HPC GPU-based clusters proposed by Kousha in [32]. Distributed ML training performance is also analyzed in [28] and [5], and communication however there is no complete framework that allows reproducible performance analysis of ML workloads on modern distributed systems. With SURFBoard, we offer a common ground for all these types of analyses to be performed in comparable settings and in a reproducible fashion.

**ML workflows and orchestration.** As depicted in Figure 1, Machine Learning workflows are composed of elements spanning a broad software stack, going from efficient GPU kernel execution, CPU-GPU work partitioning, efficient storage access, and multi-node orchestration. The interaction between these elements often leads to complex systems producing results that are very challenging to reproduce, both from a numerical (i.e. model accuracy) perspective [27], as well as from a performance perspective [7]. Typical training workflows, such as the computer vision one presented

in this work, include stages such as data preprocessing and augmentation [13], hyperparameter tuning [2, 36], or model interpretation [51]. The high computational complexity of the training process often requires the workflow to be executed in a distributed fashion, adding additional dependencies to distribution mechanisms such as Horovod [43] or PyTorch Distributed [35] and orchestration tools such as Kubernetes or SLURM. SURFBoard offers practitioners an easy-to-use and configurable framework for gathering performance data from all the components of the training workflows.

**ML/DL benchmarks.** Benchmarking large-scale system behavior under diverse workloads like HPC and big data is a well-studied topic. Recently, with the highly increased interest in ML and DL workloads, several benchmarks [7, 14, 21, 30, 38, 54] emerged to cover this need. However, due to the relatively early days of the field, none of them have emerged as a community- or industry-wide de-facto benchmark. It is also unclear at the moment how easy it is to port these benchmarks to all possible infrastructure that runs ML/DL code. Our approach helps in this sense by being able to build reproducible instances of these benchmarks to run on many types of large-scale infrastructure. Moreover, our containerized approach would ensure an even playing field (i.e., common software infrastructure) to cut down on technology- and software-induced performance differences.

Overall, SURFBoard presents a more holistic approach at achieving performance reproducibility in large-scale systems when running ML workloads. Even though SURFBoard is related and contains technologies from each of the aforementioned categories, SURFBoard is more than the sum of its components, as it is the first enabler of reproducible ML performance analysis at scale.

## 8 Conclusion

Many large-scale software systems suffer from poor performance reproducibility. Machine learning training workflows are no exception as their performance analysis is largely an expert-driven task, tightly-coupled to the underlying physical and software ecosystems. This behavior hinders productivity, knowledge sharing, and overall the notion of achieving energy efficiency.

We presented our approach at supporting reproducible performance analysis for machine learning workflows through a containerized framework. This framework is able to run on many container-ready types of infrastructure, such as HPC clusters and even clouds. Moreover, it is able to gather performance results at arbitrary levels in the software stack and is extendable such that more experienced users are able to add custom analyses.

We validated our framework through an empirical evaluation on two GPU-enabled, large-scale production-based HPC clusters, with different software stacks. Our analysis shows that our framework is portable and is able to gather perfor-

mance data ranging from high-level MPI metrics, down to FLOP efficiency for CUDA kernels, as well as kernel-level data for each processing batch. For future work, we plan to extend our framework with more types of analysis tools, implement multiple types of workloads regarding state-of-the-art benchmarks, as well as evaluate them on more types of large-scale infrastructure.

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
