# OpenReview forum: "SURFBoard: Reproducible Performance Analysis for Distributed Machine Learning Workflows"
_JSYS/2021/Mar_Papers — Revise_

### Official Review · AnonReviewer3 · 2021-03-29
**A valuable contribution to the community but requires some clarification about the implementation effort.**

**Decision:**

Weak accept: good paper with flaws that can be fixed in three months

**Review:**

Summary:
This paper proposed a customizable container-based profiling framework for ML work's reproducible performance analysis. However, the mentioned core-hours the authors invested on each of the clusters for data gathering and analysis make me wonder about how much time it will require for a developer/researcher to perform the performance analysis of a new ML workflow.

Pros:
(1) This work is precious to progress the community and ensure reproducibility. I especially appreciate the effort to develop a modular and highly configurable framework.
(2) The proposed framework can determine the bottlenecks of an ML workflow pipeline which will significantly help the community.
(3) The proposed analysis approach combines the approaches proposed by various previous works to provide a more holistic approach.
(4) I appreciate the rich related work section that clearly states the difference with the proposed framework.
(5) The high-level goals mentioned in Section 3.1 are well thought and clearly show the framework's purpose.
(6) The evaluation of the two clusters is thoroughly performed and considers a wide range of parameters.
(7) The manuscript is very well written and easy to follow.


Cons:
(1) I am concerned about how much effort is required by the researcher/developer to utilize this framework for another ML workflow.
(2) This manuscript does not mention whether containerization affects performance? Is there a trade-off between the reconfigurability and the efficiency of the framework?
(3) I am curious about how and why the authors chose the parameters in Table 1.
(4) The performance analysis is done on only one model and dataset. I am curious how the chosen model, along with the cluster configuration, affects the bottleneck. Point 5 of Section 5 states that the importance of preprocessing worker and batch size varies for Lisa and Cartesius. Though section 5.4 mentions that the under-provisioning of CPU compute to GPU compute may cause this difference, I expected such a performance analyzer to provide concrete reasoning behind each bottleneck.
(5) This manuscript does not mention whether this framework applies to clusters that share resources.

Comments to the Reviewer:
Thank you very much for submitting your work to JSys. I appreciate the paper and think that it is a valuable contribution to the community. However, I am concern about the required effort to use this framework for other models and clusters. Following are some of the concerns I have, and I believe that addressing this will make this work stronger and more valuable to the community.

(1) It will help the community if the authors explicitly mention how much effort a developer/researcher needs to invest to use this framework.
(2) I would love to see the overhead due to containerization in both the system—a discussion or experiment showing how the trade-off between reconfigurability and performance will be highly valuable.
(3) A discussion on how the parameters in Table 1 are selected will help the community.
(4) I will appreciate an experiment/discussion on how different models and cluster configurations affect the bottlenecks. Also, I will enjoy a discussion of how this analyzer can provide the reasoning behind the bottlenecks.
(5) A discussion on how this framework applies to shared resourced clusters.

Typos:
(1) In the last line of the first paragraph of Section 3.1, the number of sub-objectives should be four instead of three

**Expertise:**

Follow the literature closely, last published 5+ years ago

**Useful:**

yes

---

### Official Review · AnonReviewer4 · 2021-04-06
**This paper is not well written or well motivated to justify and demonstrate why the proposed method could achieve reproducibility. The review does not think it meets the acceptance requirement.**

**Decision:**

Weak reject: interesting papers with flaws, not sure if they can be fixed in three months

**Review:**

#### Significance:

##### Pros:
This paper studies an increasingly important problem in open science: reproducibility. Instead of studying how to achieve reproducible results, this paper studies reproducible performance analysis, focusing on ML application. Therefore, this paper proposes a new performance analysis framework called SURFBoard. The key contribution is to implement and evaluate this framework. The paper argues that a large amount of implementation efforts have been made for the development of this framework. The paper evaluates SURFBoard on one deep learning application Resnet on two large-scale GPU clusters for performance analysis to demonstrate reproducibility.

##### Cons:
The paper does not do a good job in motivating and justifying why the proposed approach can make a significant difference compared to prior work, especially no insights on how this paper achieves reproducibility. What would happen if performance analysis does not have reproducibility? How big the difference would be? These questions are not discussed or empirically evaluated.

#### Clarity:
##### Pros:
This paper describes the implementation details such as what hardware and software tools are used.

##### Cons:
This paper is not clearly written for several reasons:
1. The introduction is confusing and does not deliver important information. What is the definition of reproducible performance analysis? What is the difference between performance analysis and reproducible performance analysis? Why do people care reproducible performance analysis? What's the relationship between performance analysis and reproducibility? The paper is not clearly justify why analyzing performance of ML workflows can help reproducibility.
2. "performance analysis of ML" or similar phrases appear multiple times in the paper. The reviewer is not sure what it means. What is ML in this context? ML systems, ML algorithms, or ML models? Also, the paper uses ML workloads/algorithms/models interchangeably, which is actually confusing. The reviewer guesses that the authors mean the same things, but the meanings of these terms are subtlety different.


#### Quality:
##### Pros:
The paper conducted extensive experiments of the chosen DL application on the chosen hardware.

##### Cons:
1. It is not enough to evaluate only one workload in only one domain with one software in system paper evaluation.
2. There is no baseline for comparison. In a rare case, it is fine if the proposed work studies a new problem that no prior work has studied before. But, this paper studies reproducible performance analysis. What kinds of metrics in the evaluation can reflect the reproducibility? Actually, it goes back to the initial question: what is the definition of reproducible performance analysis? How do you justify that the proposed approach has reproducibility? Are the metrics good enough to compare reproducibility? The reviewer cannot find the answers in the evaluation.



**Expertise:**

Actively publishing in this area

**Useful:**

yes

---

### Official Review · AnonReviewer1 · 2021-04-07
**Review of the paper "SURFBoard: Reproducible Performance Analysis for Distributed Machine Learning Workflows"**

**Decision:**

Weak accept: good paper with flaws that can be fixed in three months

**Review:**

### Summary

This paper presents SURFBoard, a framework for data scientists to perform detailed performance analysis of distributed machine learning pipelines. SURFBoard packages different profilers and ML tools into a container image and provides visualization notebooks to show and study the profiling results. The authors demonstrate the type of analysis that can be performed with SURFBoard by training a ResNet50 model on two different GPU clusters, collecting profiling metrics, and visualizing and analyzing the results.

### Strengths

- Tool for automated, detailed performance analysis for ML pipelines is very useful
- Tool allows automated exploration of infrastructure parameters
- Evaluation has been done on two real-world, large-scale GPU clusters

### Areas for improvement

- The exact meaning of "reproducible performance analysis" and how that is supported by SURFBoard is not clear enough
- The description of SURBoard is missing several important details
- The paper is missing a comparison to existing ML model management frameworks such as MLflow or Pachyderm

### Comments for authors

Thank you for submitting your work to JSys! I think your paper is filling an important gap in the currently available ML tooling space, i.e., that of providing automated, detailed performance profiling as part of ML pipeline execution. Being able to, as you say "gathering performance data from all the components of the training workflow" is useful to debug and optimize pipelines. This is particularly true in the case of distributed training, where collecting such metrics becomes even harder. I liked the fact that SURFBoard allows automated exploration of infrastructure parameters. This is a nice feature that can be provide useful guidance to users, who would like to, e.g., retrain a model and need to size/estimate their resource requirements. Overall I think the community would benefit from your tool.

However, I think there are three main issues in the current paper that should be addressed before publication:

1. While reading the paper, it never was fully clear to me, what exactly "reproducible performance analysis" means and why the reproducibility aspect is particularly important for ML. Reproducibility in your case could mean a variety of things. For example, it could mean that individual experiments have reproducible performance across runs or that somebody else could use SURFBoard to train a model on a different cluster and achieve the same performance. It could also mean that SURBoard can be used the same way in different environments (which would be more like the portability that you also mention in the paper) or that it collects and makes available all the necessary information (e.g. hardware, hardware configuration, software configuration, etc.) that is required to reproduce the performance from the initial experiment. Given these different potential meanings, it is important to clearly and early on define, what you mean by reproducibility in the context of SURFBoard.

I was also missing a bit stronger motivation for why performance reproducibility is crucial for ML workloads. I completely agree that detailed performance profiling and analysis is useful and necessary but I wasn't sure about the reproducibility part of it. You mention in the introduction that it is important for productivity, model and knowledge sharing, and energy efficiency. While the energy efficiency part made sense, I wasn't sure about the other two parts. How does performance analysis/reproducibility help productivity or model sharing? Is it because it helps to optimize models to make training more efficient? I think it would be good to elaborate a bit more on this point to strengthen the motivation. This issue might also resolve itself once the reproducibility meaning has been clarified.

2. I felt that the description of SURFBoard was incomplete. After reading the paper, I didn't have a good sense for what I would need to do to deploy and use SURFBoard in my environment. In terms of system setup, some questions I had where: What are the different components that the system consists of, where do they need to be deployed, are they containerized or are infrastructure changes required? In terms of usage, I was wondering: What do I need to do to collect performance profiling metrics with SURFBoard, does it require code changes to my existing pipelines and how many, can I configure the collected metrics? Some of this information is missing from the paper while some is scattered across it. I think it would be good to have one section to describe in detail the architecture and the programming model of SURFBoard to give readers a better feeling for how they could use it.

Related to this, it would also be good to show an example for the effort it takes to extend SURFBoard to a different ML framework, profiling tool, data loader, etc. In Section 6 you say that extensibility is difficult. That is fine to have as an initial limitation but it would be good to quantify that in some way, e.g., by lines of code changes required or time it took to make the changes. This would give users a better idea of *how* difficult adaptation would be and help them to decide whether it is feasible to deploy SURFBoard in their specific environments.

3. Finally, I was wondering, whether you considered the option of integrating SURFBoards's profiling capabilities with any of the existing ML management frameworks such as MLflow, Determined AI, Pachyderm, Polyaxon, etc. Those frameworks already allow for reproducible training, hyperparameter exploration, and some even collect basic resource utilization statistics. Would it be possible to extend those frameworks to provide SURFBoard's profiling mechanisms on top of the other features? Why would that be difficult? Some of those questions might be answered once comment 2 has been addressed but I would still like to see a more explicit discussion of "if and how" or "why not" SURFBoard can be integrated with such tools. A distinction to those tools in the related work should also be added.

### Other Comments/Clarifications

- The last/first paragraph on page 1/2 wasn't fully clear to me. What are the exact reasons that new, complex ML pipelines make reproducible performance analysis more difficult? Is it because they consist of more components and have more dependencies so profiling needs to happen at many different points in the stack?

- You mention the visualization notebooks that are part of SURFBoard. How general are they? How easily can they be adapted to other metrics?

- Can you better indicate the two sections of host and containerized code that you're referring to in Figure 2? I wasn't sure where exactly the boundary was.

- What did the set up of the NVTX profiling infrastructure entail? Why was it difficult?

- In Section 3.1 you mention that your goal is to construct a performance model for performance extrapolation. Is this referring to the cost model of the volume of MPI_Allreduce messages? I was expecting something more general for this, maybe one (or several) model(s) for different metrics, integrated into SURFBoard. Would that be feasible? Are you planning to provide such a set of performance models?

- Can you clarify what is the difference between workers and nodes?

- Can you provide more details on the NVTX annotations? Who needs to add those annotations? Where do they need to be added?

- I found Figures 3 and 4 a bit confusing. First, it seemed that on the LISA cluster (Fig. 4), scaling wasn't actually linear but rather started to flatten after 4/16 nodes/GPUs. Also, I think I got confused by the way you measure duration and I didn't fully understand why duration increases with more GPUs. Shouldn't it still be going down, despite the additional network traffic, as you're doubling the processing power? Or is the network the main bottleneck?

- The analysis of the network traffic felt a bit thin, given the amount of graphs associated to it (Figures 5-10). Could you comment a bit more on the differences you observed between the different clusters? Was there anything interesting you found from looking at the network metrics? If not, some Figures (e.g. 7 and 8) could be removed to make space for some extra content. Additionally, the captions of those Figures are also repetitive.

- I was wondering, if the data in Figure 11 could be broken down further, e.g., what portion of the backward pass was network traffic, CPU/GPU data exchange, etc.? Do you think that is possible?

- Could you add the GPU statistics from the LISA cluster to Table 4 or was there a specific reason why those haven't been added?

- I liked the Pyprof-based kernel analysis, that seems to be really useful to identify major bottlenecks.

- In the Related Work, you mention that you adhere to reproducibility methods for achieving reproducible performance. Is that something that is integrated in SURFBoard or is that currently something the user needs to do manually?

### Nits

- page 1: "This is because OF the sheer complexity [...]"
- page 2: "[...] container-enabled large-scale HPC infrastructureS."
- page 6: "It is important to note that THE DL model described above is used as an example for THE validation study and to showcase THE capabilities of the framework [...]"
- page 8: "FigureS 6 and 5 presents [...]" -> present (without s)
- page 8: Resnet50 -> ResNet50

**Expertise:**

Actively publishing in this area

**Useful:**

yes

---

### Official Review · AnonReviewer2 · 2021-04-14
**Summary: Major revisions**

**Decision:**

Weak accept: good paper with flaws that can be fixed in three months

**Review:**

Summary of the paper: This paper proposes SURFBoard a container-based solution for performance evaluation of ML workflows. They show on two different use cases that the container can be used to generate results for a variety of performance questions.
The objective is to make it easier for users to share performance and efficiency characteristics of ML workflows.


Quality:

Strengths:
#1. The uses cases are comprehensive and methodology for doing experiments is well-explained.
#2. The authors have done a comprehensive survey of related work. The issues are in the research problem and proposed solution

Weakness:
#1. Is the research problem with "reproducing ML workflows", or is it "analysis of ML workflow performance" or is it "evaluation of ML workflow performance" or is it "profiling of ML workflow performance"? Given the proposed solution, it seems the problem is of challenges in "sharing ML workflows and their performance results".  Sharing a workflow raises the question of what to share and the container-based solution is an approach for improved sharing. In that regard, sharing is conflated with reproducing, analyzing, evaluating and profiling.

#2. It is not clear what is the role of reproducibility beyond portability via containers, which is a limited use of definition of reproducibility.
In particular, how using a container, which is for isolation, aids reproducible performance? If a performance result differs on a H/W it will differ irrespective.
further if application has library conflict with other libraries in the container, SURFBoard will not run.

#3. It would help if the authors describe in more detail what are the challenges behind running this framework, since they claim "performance analysis framework is able to run and achieve significant results on multiple types of infrastructure.". Is it difficult to get this framework to run because contained software has dependency conflicts and is notorious to port, or is the application using some external libraries, or if multiple types of infrastructure do not support containers or these libraries?


#4. The proposed solution is not compared with alternate approaches for sharing such as
virtual machines (i.e., SageMaker and similar offerings from Azure) and application virtualization, which can lead to either lacking jitters in some cases, or a more light-weight solution.


Clarity: The paper lacks clarity of exposition.

#1. Some concepts such as performance evaluation and analysis are overlapping and have not been distinguished clearly.

#2. The paper, in its current form, is at best vague on the specific research question that it aims to address. See #1 in Quality.

#3. The paper is unclear as to why their solution suffices?

Originality: Building containers of complex software stacks is a recent trend to improve devops and speed deployment. The paper shows that several HPC libraries can be part of a singularity container and this container can be successfully used to generate complex performance results often needed for HPC workflows.

If the paper focuses on the specific research problem, originality will be more highlighted.

Significance: The work is significant as specific containers that ease sharing of complex software must be made known to the broader community.


**Expertise:**

Actively publishing in this area

**Useful:**

yes

---

### Meta-Review · Area_Chairs · 2021-04-15

**Recommendation:** Revise
**Confidence:** 5

**Metareview:**

Dear authors,

Thanks a lot for submitting your work to JSys!
After considering all reviews and discussing among the review board, we have concluded that it is not possible to properly judge the value and uniqueness of SURFBoard, mainly due to the way is currently written. We would like to give you the chance to fix this during revision. However, if after revision, it is clear that it does not provide significant value to the community, the paper will be rejected.

In general, reviewers found that SURFBoard is not properly contextualized with respect to the state-of-the-art ML tooling landscape, both from the academic and "mainstream" perspective. Other important aspects, given that this is a "Tool/Benchmark" submission, are those related to the usability of the framework, as well as its generality. This information is of great importance for determining the value and uniqueness of the tool.

Reviewers have done an excellent job at providing constructive and detailed feedback, so please make sure you address all of it before you send the revised version. You can find instructions on how to prepare it at https://escholarship.org/uc/jsys/cfp.

Please let us know if you have any questions.

---

### Decision · Program_Chairs · 2021-04-15

Revise